# A Renewable Biosensor Based on LNA-Aptamer Hybridization for the Detection of *Salmonella enteritidis* in *Penaeus vannamei*

**DOI:** 10.3390/molecules28010450

**Published:** 2023-01-03

**Authors:** Zhihua Li, Ling Zhao, Qian Wu, Xue Zhang, Xiaowei Huang, Jiyong Shi, Xiaobo Zou

**Affiliations:** 1School of Food and Biological Engineering, Jiangsu University, Zhenjiang 212013, China; 2International Joint Research Laboratory of Intelligent Agriculture and Agri-Products Processing (Jiangsu University), Jiangsu Education Department, Zhenjiang 212013, China

**Keywords:** locked nucleic acids, aptamer, *Salmonella enteritidis*, renewable electrochemical sensor

## Abstract

*Salmonella enteritidis* (SE) is an important factor causing foodborne disease, and electrochemical sensors have drawn much attention for SE prevention and detection due to their many advantages. A renewable electrochemical sensor using specially designed locked nucleic acids (LNA) as linkers for the detection of SE was proposed to improve the reusability and reproducibility of biosensors. One end of the LNA was designed as an anchor to attach to modified electrodes through the sulfhydryl group; the other end was used to match with a short segment of SE aptamers, which will allow for the convenient renewal of occupied aptamers by raising the temperature. Results revealed that the manufactured biosensor had good stability, reproducibility, and selectivity in addition to a linear range of 6 × 10^1^–6 × 10^5^ CFU/mL and a limit of detection (LOD) of 20.704 CFU/mL. The recovery rate of SE for the real sample varied from 98.84% to 134.82% without exceeding 16.27% in the relative standard deviation (RSD). The proposed biosensor appears to be a promising tool for foodborne pathogen detection.

## 1. Introduction

*Salmonella enteritidis* (SE) is a foodborne pathogen hosted by both humans and various animals, which could cause the death of livestock, resulting in serious economic losses [1,2] and seriously endangering human health by contaminating livestock products [3]. Consequently, the creation of quick and precise detection techniques is crucial for the management and prevention of SE disease [4].

In recent years, several techniques have been used to detect SE. The traditional cultural method is reliable but has obvious limitations as it is time-consuming and labor-intensive [5]. The enzyme-linked immunosorbent assay (ELISA) is highly specific but has stringent operational requirements and insufficient sensitivity [6,7]. Polymerase chain reaction (PCR) is sensitive but poorly reproducible and needs expensive equipment [8,9]. Electrochemical methods have the advantages of being fast, simple, low cost, and highly sensitive when compared to the above-mentioned methods [10]. Therefore, there is a strong interest in developing electrochemical biosensors for the detection of SE.

Biological recognition elements are the most important part of electrochemical sensors, which could be enzymes, antigens/antibodies, or nucleic-acid aptamers [11]. The low cost, stability, and ease of synthesis of nucleic acid aptamers as well as the designability and flexibility of their structures provide more possibilities for the development of new electrochemical sensors [12]. Due to their considerable benefits, nucleic acid aptamer-based electrochemical sensors have shown considerable promise for the detection of SE. A reduced graphene oxide–titanium dioxide nanocomposite-based aptamer sensor was created by Muniandy et al., with a detection limit of 10 CFU/mL for SE, and a glassy carbon electrode (GCE) was modified by Ma et al. with graphene oxide and gold nanoparticles to link the aptamer with a detection limit of 3 CFU/mL [13,14]. However, the fabrication of these sensors is very tedious, especially for the modification of electrodes. It is necessary to develop a renewal method to enhance the reusability and reproducibility of aptamer sensors.

Locked nucleic acids (LNA, also known as bridged nucleic acids) have a methylene bridge in their structure that reduces the flexibility of the ribose and improves the stability of the phosphate backbone [15,16,17]. This structure results in each LNA substitution raising the melting temperature (T_m_) by as much as 10 °C [15,18]. LNA can hybridize specifically with nucleic acid sequences and avoid mismatched strands more effectively [19]. Moreover, they are resistant to nuclease degradation and are non-toxic in vivo [20].

In this study, LNA was used as linkers to create a renewable electrochemical biosensor for the detection of SE. One end of the LNA was designed as an anchor to attach to modified electrodes through the sulfhydryl group; the other end was used to match with a short segment of SE aptamers, which will allow the convenient renewal of occupied aptamers by raising the temperature. The proposed renewable electrochemical sensor showed high sensitivity, strong selectivity, and remarkable stability according to the experimental results, and it was effectively used with the actual samples.

## 2. Results and Discussion

### 2.1. Working Principle of the Electrochemical Biosensor

The renewable biosensor’s operating system is depicted in Figure 1. Three main parts make up the biosensor: (1) A biometric component, known as an aptamer, that specializes in the detection and capture of SE was expanded with 18 nucleic acid bases (GAAGGGCTTTTGAACTCT) to connect with an LNA; (2) the LNA serves as a link between the biometric element and the signal converter, which might increase the base recognition and phosphate backbone stability significantly; and (3) the GCE electrode was modified with gold nanoparticles to function as a transducer for signal.

When SE was used as a detection target in the sample, the aptamer would specifically bind to the SE surface receptor through its secondary structure. Then, the detection of different concentrations of SE in the sample was achieved by measuring the impedance value of the electrode due to the electron shielding effect of the cell membrane surface, which hinders the electron transfer. The aptamer–bacteria combinations were eliminated by raising the temperature to 80 °C after their identification. The stronger phosphate backbone of LNA and its tight binding to the electrode made the LNA/Au/GCE electrode repeatable.

### 2.2. Characterization of Renewable Biosensor Based on LNA-Aptamer Hybridization

Cyclic voltammetry (CV) and electrochemical impedance spectroscopy (EIS) were used to investigate the electrochemical behavior of the renewable biosensor during the preparation process [21]. Electrochemical detection was carried out in a 5 mmol/L electrolyte solution of [Fe (CN)]^3−/4−^, with [Fe (CN)]^3−/4−^ being a redox-reduction pair [22]. In this study, CV experiments and EIS experiments were carried out in a 5 mmol/L electrolyte solution of [Fe (CN)]^3−/4−^ consisting of five different modified electrodes of (a) bare GCE, (b) Au/GCE, (c) LNA/Au/GCE, (d) Apt/LNA/Au/GCE, (e) SE/Apt/LNA/Au/GCE (Figure 1). The reversible peak of Au/GCE (Figure 1A, curve b) was a little higher than the bare GCE (Figure 1A, curve a) and its charge transfer resistance was less than the transfer resistance of the bare GCE (Figure 1B, cure b and cure a, respectively), indicating that gold nanoparticles deposited on the electrode surface promote conductivity of the electrode. The connection of Au/GCE with LNA (Figure 1A,B, curve c) resulted in a decrease in the reversible peak and an increase in the charge transfer resistance (R_ct_). After SE incubation with apt/LNA/Au/GCE to obtain SE/Apt/LNA/Au/GCE, the electrode had a lower reversible peak (Figure 1A, curve e), and R_ct_ increased significantly (Figure 1B, curve e), indicating that the SE was caught due to the bacterial cell membrane’s electrical shielding action.

Additionally, the SEM description corroborated this (Figure 2). As shown in Figure 2A, the Apt/LNA/Au/GCE electrode exhibited a compact and rough appearance (Figure 2A). After incubation with SE and rinsing with PBS, rod-shaped translucent bacteria were detected (Figure 2B), indicating that SE bacteria were successfully identified.

### 2.3. Optimization of Renewable Biosensor Performance

EIS technology was used to investigate the optimization of renewable biosensor performance. LNA can hybridize with aptamers and its concentration may affect the sensitivity of the sensor. The R_ct_ of Apt/LNA/Au/GCE increased in proportion to the LNA concentration and plateaued at 1 µmol/L (Figure 3A). Therefore, LNA at a concentration of 1 µmol/L allows for more attachment of aptamers.

The unlinking between the LNA and the aptamer has a direct effect on the recovery of the sensor, which can be regulated by varying the temperature of the unlinking. ΔR_ct_ was discovered to significantly climb and approach a plateau around 80 °C (Figure 3B). In the following experiment, 80 °C was the optimum unchaining temperature.

Similarly, the renewable biosensor was cultured with SE to see whether the recognition time affected it. R_ct_ rose with incubation time, as seen in Figure 3C, and remained stable after 60 min. As a result, 60 min was chosen as the right amount of time for incubation in the next study.

### 2.4. Salmonella enteritidis Quantitative Detection

SE was quantitatively monitored using EIS experiments under the optimum conditions. As illustrated in Figure 4A, the R_ct_ value rose steadily as the concentration of SE increased, confirming a SE concentration-dependent mode of charge transfer resistance enhancement due to the bacterial cell membrane’s electrical shielding action [23,24]. Moreover, the peak current exhibited an excellent linear relationship with the logarithmic value of SE concentrations in Figure 4B, following the regression equation of R_ct_ (Ω) = 523.932 + 130.216 LogC (CFU/mL) and a correlation coefficient of 0.992. The low detection limit of 20.704 CFU/mL was achieved for SE, which was attributed to the recognition of SE by the aptamers. In comparison to the prior products and reported electrochemical sensors for SE monitoring, the biosensor developed in this study exhibited a high level of sensitivity.

### 2.5. Stability, Repeatability, and Selectivity of the Renewable Biosensor

The renewable biosensor was stored in a PBS solution (25 °C, 0.01 mol/L, pH = 7) for four weeks. Then, the stability of the renewable biosensor was investigated by comparing the weekly variation of the EIS response to SE. The R_ct_ of the renewable biosensor exhibited nearly no variation in Figure 5A. The renewable biosensor still retained 93% of the electrochemical signal after being stored for four weeks, demonstrating that the constructed sensor had enough stability. Another critical factor to consider when evaluating sensors is repeatability.

The repeatability of the renewable biosensor was evaluated by comparing the EIS response to SE of five Apt/ LNA/Au/GCE electrodes prepared under the optimal conditions. The R_ct_ of 6 × 10^4^ CFU/mL SE detected by the biosensors showed very little difference between each other in Figure 5B. With a RSD of 4.085%, this suggests that this biosensor had good repeatability.

The selectivity of the renewable biosensor was assessed by comparing the difference in the resistance of the renewable sensor to the charge transfer of three bacteria solutions (*Staphylococcus aureus*, *Escherichia coli*, and *Lactobacillus Plantarum*) at the same concentration. As shown in Figure 5C, the ΔR_ct_ of the renewable biosensor for SE had a huge gap compared to that of the other bacteria, moreover, the ΔR_ct_ of the renewable biosensor for other bacteria was not significantly different from the control (PBS solution), suggesting that the renewable biosensor possesses obvious selectivity.

### 2.6. Renewability of the Renewable Biosensor

After each measurement, the detected electrode was immersed in 0.01M PBS buffer (80 °C) for 8–10 min, then incubated with 5 µL of a 100 nmol/L SE aptamer, and the unbound SE aptamer was washed off to reconstruct the sensor. The detection efficiency was almost restored to the initial level. Repeating the above operation, the detection efficiency of the seventh measurement was only reduced to 88.99% of the starting value (Figure 6). In other words, within the range of signal loss not exceeding 10%, the electrode of the renewable biosensor could be effectively regenerated five times. On one hand, the decline in detection efficiency can be attributed to the self-assembled LNA/Au/GCE falling off during the unchaining process. On the other hand, the recovery of detection efficiency may be attributed to the modification of oligonucleotides by LNA to improve the stability of the phosphate backbone. Recycling electrodes has outstanding practical value, providing a new idea for electrochemical sensor research.

As a comparison, we used DNA as a connecting element and made reproducible electrodes in the same way. After repeating the above tests and modifications, the detection efficiency was reduced to 66.52% of the starting value in the second measurement and even to 40.33% of the starting value in the fourth test (Figure 6B). A comparison of the two connecting elements showed that LNA provided a significant contribution to the regeneration performance of the electrodes.

### 2.7. Real Sample Analysis

A renewable biosensor was used to determine the different concentrations of SE in the *Penaeus vannamei* sample solution to assess its utility. As can be seen from Table 1, the recovery rate of SE for the *Penaeus vannamei* sample varied from 98.84% to 134.82% without exceeding 16.27% in the relative standard deviation (RSD). According to the results, the renewable biosensor provided satisfactory monitoring outcomes to identify SE in the actual samples.

## 3. Materials and Methods

### 3.1. Materials

Potassium ferricyanide (K_3_[Fe (CN)_6_]), potassium ferrocyanide (K_4_[Fe (CN)_6_]), potassium chloride (KCl), sodium chloride (NaCl), alumina powder (Al_2_O_3_, 1.0 µm, 0.3 µm, 0.05 µm), potassium dihydrogen phosphate (KH_2_PO_4_), disodium hydrogen phosphate (Na_2_HPO_4_), sulfuric acid (H_2_SO_4_), Tris(2-carboxyethyl) phosphine (TCEP), 6-hydroxy-1-hexanediol (MCH), Tris HCl NH_2_C(CH_2_OH)_3_ … HCl, and chloroauric acid (HAuCl_4_) were obtained from Sinopharm Chemical Reagent Co. Ltd. (Shanghai, China). As a washing buffer, phosphate-buffered saline (PBS, pH 7.0) containing 2.68 mM KCl,137 mM NaCl, 1.76 mM KH_2_PO_4_, and 4.02 mM Na_2_HPO_4_ was utilized. All chemicals were of analytical reagent grade and solutions were prepared using ultrapure water. *Salmonella enteritidis* (CICC:21482) was purchased from the China Center of Industrial Culture Collection (Shanghai, China).

Without changing the aptamer’s secondary structure, the locked nucleic acid (LNA) was created to achieve a more stable phosphate backbone (the LNA and aptamer were unchained and the LNA characteristics held steady). The SE aptamer was chosen following a thorough examination of prior research. Nucleotide sequences with 18 bases were modified at the 3′ end of the aptamer, and they can be complementarily paired with the LNA. The synthesis of aptamers and LNA was entrusted to Shanghai Biotech (Shanghai, China). Nucleotide sequences of the SH-LNA and SE aptamers are listed in Table 2.

### 3.2. Equipment

The electrochemical measurements were performed by a CHI 660E electrochemical workstation (Shanghai Chenhua Instrument Co. Ltd., Shanghai, China). In a three-electrode system, a platinum wire electrode, an Ag/AgCl electrode, and a glassy carbon electrode (GCE) were used as the counter, reference, and counter electrodes, respectively. All electrodes were purchased from Sigma-Aldrich (Sigma-Aldrich Company, Shanghai, China). Scanning electron microscopy (SEM) images of the electrodes were recorded by a Hitachi S-3400N (Shanghai, China). The culture and dilution of SE were carried out in a shaker (Taicang Qiangwen Experimental Equipment Co. Ltd., Taicang, China) and on a super-clean bench (Qingdao Haier-Medical Co. Ltd., Qingdao, China). Sterilization was performed in an autoclave (Shanghai Boxun Medical Biological Instrument Co. Ltd., Shanghai, China) and the aptamers were incubated in a fridge (Konka Group Co. Ltd., Shenzhen, China). A constant temperature and humidity incubator (Shanghai Lanbao Testing Equipment Co. Ltd., Shanghai, China) was used to incubate the SE.

### 3.3. Preparation of Renewable Biosensor

The diagram for the preparation of the renewable biosensor is shown in Figure 7. First, glassy carbon electrodes (GCE) polished with Al_2_O_3_ and cleaned were immersed in a 1 mg/mL solution of HAuCl_4_ (0.5 M H_2_SO_4_) for 200 s to deposit gold nanoparticles on their surface by linear scanning voltammetry (LSV) at −0.5 V. Second, 5 µL of a 1 µmol/L SH-LNA was activated with TCEP and incubated with the modified electrode for 2 h at 36 °C to obtain very powerful Au–S coordination bonds, followed by washing with 0.1 mol of PBS buffer to remove non-specific adsorption. Third, 10 µL of a 1 mmol/L 6-mercapto-1-hexanol (MCH) was dropped onto the surface of the modified electrode for 1 h to obtain a closed electrode, which was washed with ultrapure water to remove excess MCH and then dried in an enclosed container at 25 °C. Finally, 5 µL of 100 nmol/L SE aptamers were incubated with the LNA-modified electrode for 30 min at 45 °C, followed by a rinse with 10 mmol/L Tris-HCl (PH 7.4) buffer to wash off unbound SE aptamers. The SE aptamers added in this paper were all excessive. The monitoring of changes in the electrochemical reactions between the electrode modification steps was carried out using CV and EIS technology [25,26].

### 3.4. Regeneration of Biosensor

The detected electrodes were immersed in 0.01 M PBS buffer (80 °C) for 8–10 min and then quickly submerged in ice water to cool down so that the aptamer–bacteria complexes were removed and the LNA probes were retained on the electrode surface. The regenerated electrodes were analyzed using EIS measurements.

### 3.5. Culture and Counting of Bacteria

Glycerol-conserved strains were inoculated onto the surface of the solid media for 24 h at constant temperature and humidity. A single colony was then picked and incubated in a liquid medium with shaking for 5 h in a shaker to acquire active bacteria. Then, the bacterial solution was taken and the absorbance was measured at 600 nm. The enriched bacteria were centrifuged at 8000 rpm for 2 min at 25 °C and the supernatant was discarded. The precipitate was washed three times with 0.01 M PBS (pH 7.0) solution and resuspended in 0.01 M PBS to acquire the microbiological suspension. The resulting solution was diluted by 0.01 M PBS, and 100 µL of 10^−7^, 10^−8^, and 10^−9^ dilutions of SE were coated in a sterile solid medium with three parallels for each gradient and placed in an incubator at 37 °C for 24 h [27]. The average number of colonies in each gradient was calculated to give the concentration of SE (CFU/mL).

### 3.6. Electrochemical Measurements

A three-electrode system (glassy carbon electrode, Ag/AgCl electrode, and platinum wire electrode) was used for the detection of SE. First, the modified GCE was incubated with SE suspensions (10^3^–10^7^ CFU/mL) at 37 °C for 60 min until the bacteria reacted sufficiently with the aptamer on the electrode. After being rinsed with ultrapure water, the electrode (sensor + SE) was immersed in a 5 mmol/L electrolyte solution of [Fe (CN)]^3−/4−^ for EIS testing. The following was established as the EIS measurement: Initial voltage: 0.23 V; amplitude potential: 5 mV; amplitude: 5 mV; frequency range: 0.1–100,000 Hz. The same method was used to perform specific tests on *Staphylococcus aureus*, *Escherichia coli,* and *Lactobacillus Plantarum*.

### 3.7. Recovery of Salmonella enteritidis in Penaeus vannamei

This experiment was conducted using *Penaeus vannamei* as the actual sample for the SE test. The treatment of *Penaeus vannamei* and the preparation of its sample solutions were prepared according to GB/T 4789.20-2003 (Chinese national standard). Briefly, 25 g of abdominal tissue from *Penaeus vannamei* was crushed and added to 225 mL of saline. Then, the solution was centrifuged to obtain the supernatant. Finally, 900 µL of supernatant was added to 100 µL of different concentrations of SE to obtain a sample solution.

## 4. Conclusions

In summary, a renewable biosensor based on the complementary pairing of aptamers and modified nucleotide sequences was established. Aptamers linked to SE were released from the electrode at melting temperatures (T_m_), resulting in electrodes with a longer service life. Compared with previous biosensors based on the direct connection of electrodes to aptamers, the proposed method avoids the tedious and repetitive modification of electrodes. The biosensor has a wide detection range (6 × 10^1^–6 × 10^5^ CFU/mL) and is very selective in recognizing SE and other bacteria. SE detection was also effective in the *Penaeus vannamei* samples, demonstrating that the sensor has a lot of potential for detecting foodborne pathogens.

## Figures and Tables

**Figure 1 molecules-28-00450-f001:**
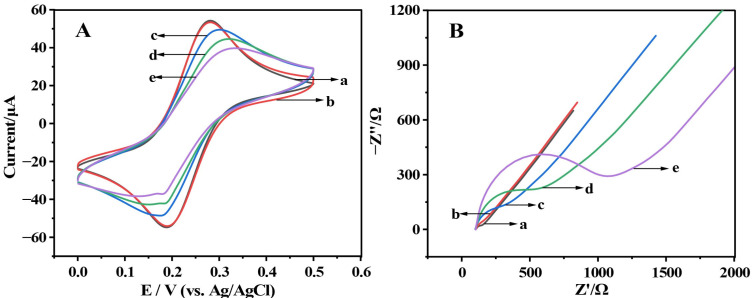
(**A**) Cyclic voltammetry curves of different modified electrodes; (**B**) Nyquist plots of different modified electrodes. (a) bare GCE, (b) Au/GCE, (c) LNA/Au/GCE, (d) Apt/LNA/Au/GCE, (e) SE/Apt/LNA/Au/GCE.

**Figure 2 molecules-28-00450-f002:**
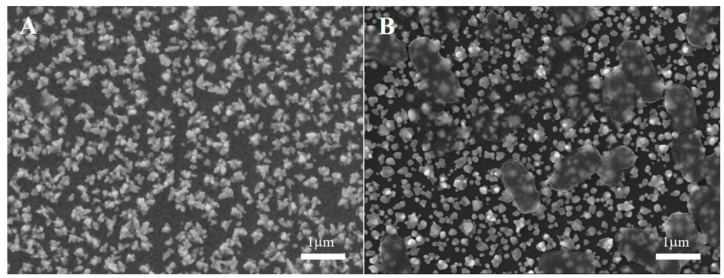
SEM images of (**A**) apt/LNA/Au/GCE and (**B**) SE/apt/LNA/Au/GCE.

**Figure 3 molecules-28-00450-f003:**
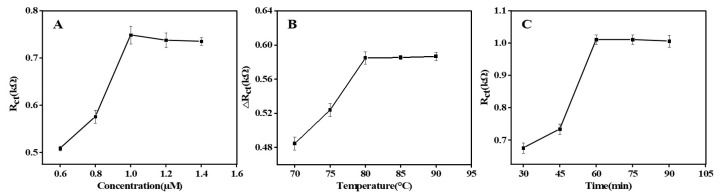
Optimization of the experimental conditions. (**A**) The concentration of locked nucleic acid. (**B**) Melting temperature of locked nucleic acid and aptamer. (**C**) Incubation time of SE.

**Figure 4 molecules-28-00450-f004:**
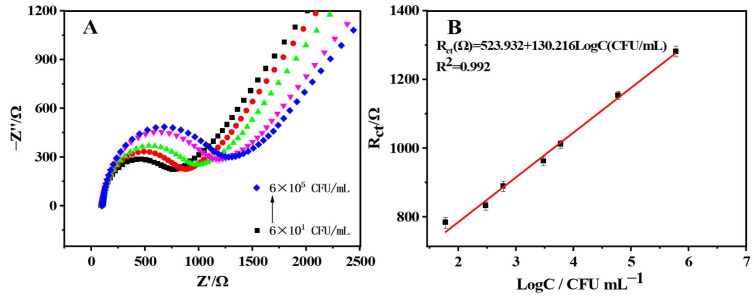
(**A**) Nyquist diagram of SE at different concentrations. (**B**) The relationship between the logarithmic value of concentration and the electron transfer impedance.

**Figure 5 molecules-28-00450-f005:**
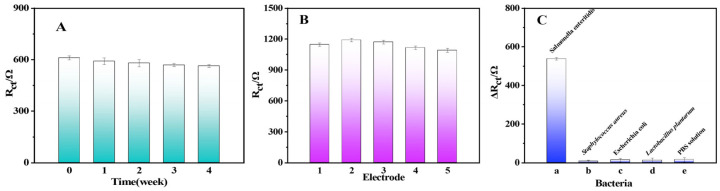
(**A**) Stability, (**B**) reproducibility, and (**C**) specificity studies of the renewable biosensor.

**Figure 6 molecules-28-00450-f006:**
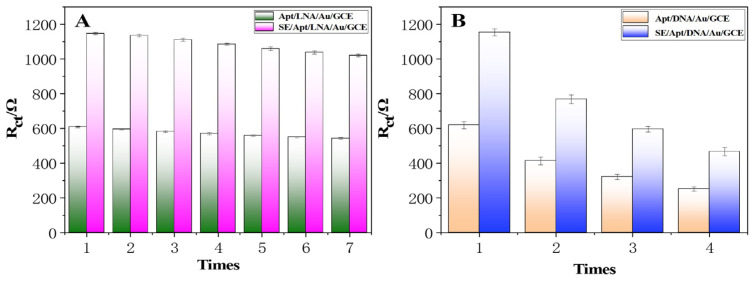
(**A**) Regenerative sensor with LNA as the connecting element. (**B**) Regenerative sensor with DNA as the connecting element.

**Figure 7 molecules-28-00450-f007:**
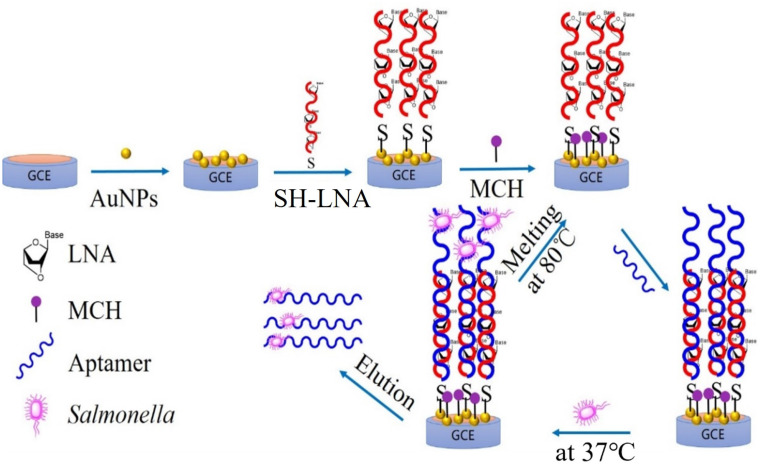
Schematic representation of the preparation of the renewable electrochemical biosensor step by step.

**Table 1 molecules-28-00450-t001:** Electrochemical biosensor based on locked nucleic acid for the detection of *Penaeus vannamei*.

Added (CFU/mL)	Measured (CFU/mL)	Recovery (%)	The Relative Standard Deviations (RSD, %)
6.520 × 10^1^	8.763 × 10^1^	134.82	7.830
6.520 × 10^2^	6.578 × 10^2^	101.20	7.110
6.520 × 10^3^	6.424 × 10^3^	98.84	16.270

**Table 2 molecules-28-00450-t002:** Details of the SH-LNA and SE aptamer.

Name	Sequences
SH-LNA	5′-HS-A^L^GAGTTCA^L^AAAGCCC^L^TTC-3′ (L: 2′-O,4′-C-methylene-(D-ribofuranosyl) nucleotides LNA)
Aptamer	5′-TTTTTGCGATCCAAGCTTCTTCAATTGGAGTGCTACCGAGATACATGGGTGGGCTCAAACAATCGTGGGCTCGCTTGATACTAGACTGCACATCTGAAGGGCTTTTGAACTCT-3′

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
