# Peer review of "A Renewable Biosensor Based on LNA-Aptamer Hybridization for the Detection of Salmonella enteritidis in Penaeus vannamei"

_molecules, 2023, doi:10.3390/molecules28010450_

Round 1

Reviewer 1 Report

There is no mention to accuracy and precision of this method. And also, it is better to confirm Salmonella enteritidis with PCR and compare CFU with real-time PCR. Figure-2,3, and 5 are not obvious to read. 

Reviewer 2 Report

In this manuscript, the authors immobilized LNA on electrodes and detected the SE bacteria. The authors made the point of regeneration of the sensor. I think it is an interesting work, but some important control experiments are missing.

1. Line 10, 'high specificity as well as sensitivity'. I don't think electrochemical sensors are always sensitive and selective. It depends on the immobilized probe molecules.

2. The quality of all the figures are very low. It's hard to see the details.

3. I don't understand why LNA was used. SH-DNA should be able to achieve the same goal, especially when you are doing regeneration, DNA should be easier to regenerate. Some comparison with DNA is needed since the main point of this paper is about the use of LNA.

4. A control aptamer is needed by mutating a few bases in the aptamer so that it cannot bind anymore. In this case, the recognition of the target bacteria should be gone. Currently, the signal could be due to nonspecific bacterial adsorption.

5. Figure 3A, the authors only studied the change of R as a function of LNA concentration. It is more important to see the effect of adding the bacterial to see what's the best LNA density.

Round 2

Reviewer 2 Report

The authors have addressed my comments.